

SciPost Phys. 1(1), 010 (2016)

# Detecting a many-body mobility edge with quantum quenches

P. Naldesi[1,2,3], E. Ercolessi [1,2] and T. Roscilde [3,4*]

**1** Dipartimento di Fisica e Astronomia dell'Università di Bologna, Via Irnerio 46, 40127 Bologna, Italy
**2** INFN, Sezione di Bologna, Via Irnerio 46, 40127 Bologna, Italy
**3** Laboratoire de Physique, CNRS UMR 5672, Ecole Normale Supérieure de Lyon, Université de Lyon, 46 Allée d'Italie, Lyon, F-69364, France
**4** Institut Universitaire de France, 103 boulevard Saint-Michel, 75005 Paris, France

\* tommaso.roscilde@ens-lyon.fr

## Abstract

**The many-body localization (MBL) transition is a quantum phase transition involving highly excited eigenstates of a disordered quantum many-body Hamiltonian, which evolve from "extended/ergodic" (exhibiting extensive entanglement entropies and fluctuations) to "localized" (exhibiting area-law scaling of entanglement and fluctuations). The MBL transition can be driven by the strength of disorder in a given spectral range, or by the energy density at fixed disorder – if the system possesses a many-body mobility edge. Here we propose to explore the latter mechanism by using "quantum-quench spectroscopy", namely via quantum quenches of variable width which prepare the state of the system in a superposition of eigenstates of the Hamiltonian within a controllable spectral region. Studying numerically a chain of interacting spinless fermions in a quasi-periodic potential, we argue that this system has a many-body mobility edge; and we show that its existence translates into a clear dynamical transition in the time evolution immediately following a quench in the strength of the quasi-periodic potential, as well as a transition in the scaling properties of the quasi-stationary state at long times. Our results suggest a practical scheme for the experimental observation of many-body mobility edges using cold-atom setups.**


# 1 Introduction

A fundamental paradigm in classical many-body physics – laying the foundations of statistical mechanics – is that of ergodicity, namely the ability of a many-body system to sample the microcanonical ensemble of states by means of its very own Hamiltonian dynamics [1]. In generic, non-random quantum many-body systems, the celebrated eigenstate thermalization hypothesis (ETH) [2,3] translates this paradigm at the level of Hamiltonian eigenstates: in the thermodynamic limit, expectation values on individual eigenstates become uniquely a function of their energy, and therefore reproduce the microcanonical averages. As the statistical ensemble is built in the eigenstates, the entanglement entropy and quantum fluctuations of macroscopic properties on extensive subsystems must obey the same volume-law scaling and mutual relationships as for the corresponding thermodynamic quantities. As a consequence, the Hamiltonian dynamics initialized in any superposition of eigenstates within a given energy window will reproduce the equilibrium values of the microcanonical ensemble after dephasing.

Disordered quantum systems offer a competing paradigm to the ETH one, the one of quantum localization, as laid down by the seminal work of Anderson [4]. The localization of single-particle wavefunctions due to disorder offers an example "in which the approach to equilibrium is simply impossible" [4]. More recent works [5–10] have convincingly shown that, in the presence of sufficiently strong disorder, an interacting quantum many-body system can display many-body localization (MBL), violating the tenets of ETH for a part or for the totality of its eigenstates. Eigenstates displaying MBL exhibit an *area law* for entanglement entropy and quantum fluctuations [11], and strong fluctuations of expectation values between eigenstates with adjacent energies. As a consequence, the dynamics initialized in a superposition of eigenstates with MBL keeps memory of the initial state and fails to equilibrate, as demonstrated in recent seminal cold-atom experiments [12–15].

A most investigated and challenging phenomenon is the transition from the ETH to the MBL regime in the Hamiltonian spectrum, and in the ensuing Hamiltonian dynamics. If the spectrum of the system is fully localized for sufficiently strong disorder, this transition can be obviously driven by the disorder strength at any energy density. Yet a generic system featuring MBL may also display a window of disorder strengths in which the spectrum is only partially localized, and spectral ranges of localized eigenstates are separated from ranges of extended eigenstates by so-called many-body mobility edges. The strongest evidence for the spectral MBL transition comes from exact diagonalization on small disordered one-dimensional quantum systems [6,7,16–26], an aspect which might prevent the accurate location of the transition point, as witnessed by the disagreement with other approaches such as the linked-cluster expansion [27]. Moreover recent theoretical work challenges the very existence of many-body mobility edges, based on rare-event arguments [28]. Another very intriguing aspect concerns

the ETH regime close to the transition, which is predicted [29,30] and numerically shown [31] to display a very rich phenomenology, characterized by a disorder- and energy-dependent slow dynamics. These aspects (namely the existence of mobility edges, the nature of the dynamics in the proximity of the MBL transition, etc.) are still awaiting an experimental verification.

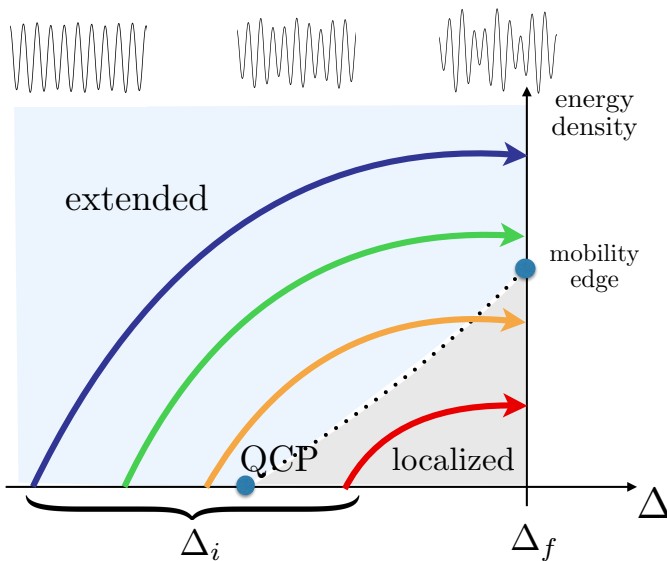

Figure 1: Principle of quantum-quench spectroscopy for the detection of a many-body mobility edge. The system is prepared in the ground state (or, more generically, a low-energy state) with a given initial depth $\Delta_i$ of the (quasi-)random potential (in this case, the secondary component of the incommensurate superlattice, depicted in the top part of the figure). The (quasi-)random potential is then quenched suddenly to a fixed final value $\Delta_f$, thereby injecting an amount of energy density in the system which is controlled continuously by the quench amplitude $\Delta_f - \Delta_i$. A sufficiently large quench amplitude may provide enough energy to cross a many-body mobility edge (which we assume to be continuously connected to a ground-state quantum critical point – QCP), and to probe it by monitoring the short-time and long-time post-quench dynamics.

Our present work aims at establishing a theoretical as well as experimental protocol to probe unambiguously the existence of a many-body mobility edge, and the very peculiar nature of the dynamics upon approaching it. Inspired by theoretical and experimental observation of MBL in incommensurate optical superlattices [12,32], we focus our theoretical study on a model of $1d$ interacting spinless fermions in a quasi-periodic potential. This model displays a localization transition at a non-trivial critical value of the quasi-periodic potential in its ground state, and a similar transition in its most excited state – as we thoroughly demonstrate numerically, making use of quantum Monte Carlo (QMC) and density-matrix renormalization group (DMRG). The ground state and the most excited state exhibit clearly different critical values for the disorder, suggesting the model as a natural candidate for the existence of an intermediate mobility edge in the spectrum. The case for a many-body mobility edge is further made by an extensive exact diagonalization study, by which we reconstruct the non-trivial dependence of the putative mobility edge on the disorder strength from the statistics of level spacings and the scaling of entropy and fluctuations.

But the strongest evidence of a mobility edge comes from an extensive study of the out-of-equilibrium dynamics. Indeed we suggest an original protocol (sketched in Fig. 1) by which

the eigenstates of a given (final) Hamiltonian $\mathscr{H}_f$ in a selected energy range are addressed by quenching the system from the ground state of a different (initial) Hamiltonian $\mathscr{H}_i$, differing from $\mathscr{H}_f$ in the disorder strength $\Delta$. A wider quench in the disorder strength from the initial to the final value amounts to populating eigenstates of $\mathscr{H}_f$ with a parametrically higher energy density. This "quench spectroscopy" method endows the study of the quasi-stationary state of the system, reached after the out-of-equilibrium dynamics, with energy resolution. In the presence of a mobility edge separating localized states at low energy density from extended states at high energy density, exact diagonalization and time-dependent DMRG clearly show the existence of two regimes of quantum quench dynamics: a regime of small quenches in which the system fails to thermalize (verifying MBL), and a regime of large quenches in which the quasi-stationary state of the system shows thermalization (revealing ETH). Most interestingly, a clear dynamical transition appears already at the level of the short-time dynamics, with small quenches featuring a logarithmically slow increase of the entanglement entropy, and large quenches featuring a power-law increase with an energy-dependent exponent. As the short-time dynamics is accessible to time-dependent DMRG studies as well as to cold-atom experiments, our protocol offers an original tool for scalable numerical studies of mobility edges beyond exact diagonalization, and, most importantly, for experimental quantum simulation.

The structure of our paper is as follows: Sec. 2 discusses the ground-state phase diagram of the model of one-dimensional interacting spinless fermions in a quasi-periodic potential; Sec. 3 focuses on the properties of the excitation spectrum and the level-statistics, entanglement and fluctuation signatures of the existence of a many-body mobility edge; Sec. 4 discusses the dynamical signatures of the mobility edge as probed via quantum-quench spectroscopy.

## 2 Ground-state and most-excited-state localization

The theoretical investigation of MBL has mostly focused on random spin chains – in particular the antiferromagnetic XXZ model in a random field [6, 7, 16, 17, 20], and the random-bond Ising model in a transverse field [19], among others. These models have the common feature that, when not protected from disorder by an energy gap, the ground state is localized for any arbitrary disorder strength – namely it displays correlations exponentially decaying over a finite localization length. On the other hand, recent experiments on ultracold atoms [12, 33–36] have demonstrated the ability to engineer one-dimensional particles immersed in a quasi-periodic potential, which, already at the level of single-particle physics, induces localization only beyond a critical strength. The minimal model to describe the physics at play is the interacting Aubry-André model, whose Hamiltonian reads:

$$\mathscr{H}(V, \Delta, \phi) = \sum_{i=1}^{L-1} \left[ -j \left( c_i^\dagger c_{i+1} + \text{h.c.} \right) + V \, n_i n_{i+1} \right] - \sum_{i=1}^{L} h_i \, n_i \, . \tag{1}$$

Here we focus on spinless fermion operators $c_i, c_i^\dagger$ defined on a chain of length $L$, in the presence of an incommensurate potential $h_i = \Delta \cos(2\pi\alpha i + \phi)$ where $\alpha$ is an irrational number and $\phi$ a random phase factor. Here we take $\alpha = 0.721$ (as in the recent experiment of Ref. [12]). Many-body physics is introduced via the nearest-neighbor repulsion term $V$. A Jordan-Wigner transformation maps this fermionic model onto a model of hardcore bosons with the exact same interaction and external potential; or onto an XXZ spin chain with anisotropy $V/(2j)$ between the coupling of the $z$ spin components and that of the $x(y)$ spin components, and further immersed in a quasi-periodic magnetic field. In the following we will set $j = 1$ and express the strength of interactions $V$ and of the quasi-periodic potential $\Delta$ in units of $j$. Throughout this work we restrict the Hilbert space to half filling, namely $N = L/2$ particles (or $(L-1)/2$ when $L$ is odd).

In the absence of interactions ($V = 0$), the single-particle Hamiltonian is well known to undergo the Aubry-André transition [37] between a fully extended spectrum for $\Delta < 2$ and a fully localized spectrum for $\Delta > 2$ – this transition has been observed in seminal cold-atom experiments Ref. [34]. The presence of interactions enriches considerably the ground-state physics: the localized phase is preserved [38], but the localization transition is shifted in a fundamentally asymmetric way between repulsive ($V > 0$) and attractive ($V < 0$) interactions. We investigated systematically the ground-state phase diagram making use of quantum Monte Carlo (QMC) simulations based on the Stochastic Series Expansion formulation [39] as well as on the density-matrix renormalization group (DMRG) [40, 41]. The QMC approach allows to investigate the one-body correlation function of the system $g(i, r) = \langle c_i^\dagger c_{i+r} \rangle$, as well as the density profiles $\langle n_i \rangle$, for chains with periodic boundary conditions with size $L = 160$ at an inverse temperature $\beta = L$, at which thermal effects become negligible. In particular the one-body correlation function is averaged over the reference point $i$, an operation which smoothens it significantly on a large system size and essentially eliminates its dependence on the specific choice of the spatial phase $\phi$. On the other hand, DMRG gives access to the entanglement entropy of half a chain

$$S_{L/2} = -\text{Tr} \left\{ \rho_{L/2} \log_2 \rho_{L/2} \right\} \tag{2}$$

where $\rho_{L/2} = Tr_{L/2} |\psi\rangle\langle\psi|$ is the reduced density matrix of one half of a chain of length $L$ with open boundary conditions. The results shown below have been obtained for a chain of length $L = 72$.

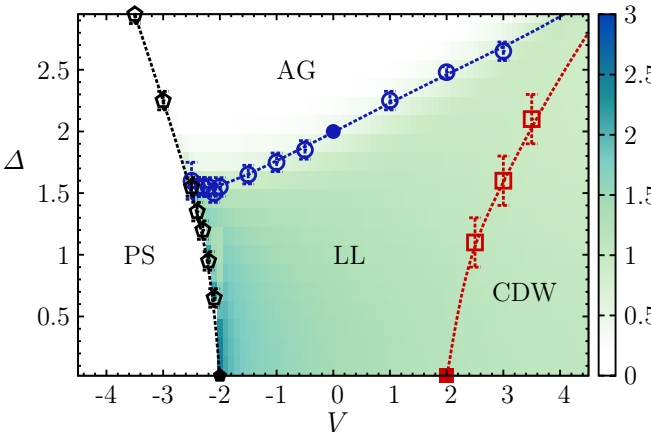

Figure 2: Ground-state phase diagram of interacting spinless fermions in quasi-periodic potential. The system exhibits the following phases: Luttinger liquid (LL), charge density wave (CDW), phase separation (PS), and Anderson glass (AG). The phase boundary have been determined via quantum Monte Carlo (see text), and the false colors indicate the half-chain entanglement entropy determined via DMRG on a $L = 72$ chain for a single realization of the phase $\phi$.

Fig. 2 shows the ground-state phase diagram of the system in the $V - \Delta$ plane as obtained via QMC. One can recognize a broad Luttinger liquid (LL) phase, characterized by algebraic correlations $g$ and a gapless spectrum; two gapped insulating regimes: a charge-density wave (CDW) phase and a phase-separated (PS) regime, both with exponentially decreasing $g$; and a gapless insulating phase, corresponding to an Anderson glass (AG), also displaying an exponentially decreasing $g$. The phase boundary between the AG phase and the PS phase is determined by the onset of droplets in the density profile $\langle n_i \rangle$. The boundaries between the

LL phase on one side, and the AG and PS phase on the other side, correspond very well to the location of a sharp drop in the entanglement entropy of the half chain, shown in false colors in Fig. 2. On the other hand the LL-CDW boundary is nearly invisible to entanglement because of finite-size effects. Indeed the CDW phase features a doubly degenerate ground state in the thermodynamic limit, and therefore a residual entropy of $\approx 1$, which is comparable to the logarithmically scaling entropy [42] $S_{L/2} \approx \frac{c}{6} \log_2(L/2)$ ($\approx 0.86$ for $c = 1$ and $L = 72$) in the LL phase for the system size under investigation.

A most important aspect of this phase diagram concerns the fundamental asymmetry between the attractive and the repulsive side. Indeed, compared to the Aubry-André critical point $\Delta_c = 2$ at $V = 0$, for $V > 0$ the critical (quasi-)disorder strength increases, due to the screening effect offered by repulsive interactions. On the contrary, for $V < 0$ the critical strength decreases due to an anti-screening effect (an attractive localized particle enhances the depth of the potential well around which it is localized). This observation is to be combined with a simple, yet crucial property of the Hamiltonian in Eq. (1): on average over the random phase $\phi$, the ground-state physics of the Hamiltonian $\mathcal{H}(-V, \Delta, \phi)$ is the *same* as that of the Hamiltonian $-\mathcal{H}(V, \Delta, \phi)$. In other words, the ground state of the attractive model $V < 0$ corresponds to the highest-energy state of the repulsive model, and viceversa. This implies that localization occurs for different critical (quasi-)disorder strengths in the ground state and in the most excited state of the Hamiltonian. If the two critical points are to be connected with a continuous transition line separating extended from localized states in the spectrum, this line has necessarily a non-trivial $\Delta$-dependence, implying therefore the existence of a many-body mobility edge. This intuition is indeed corroborated by an explicit study of the properties of the whole spectrum, as discussed in the next section.

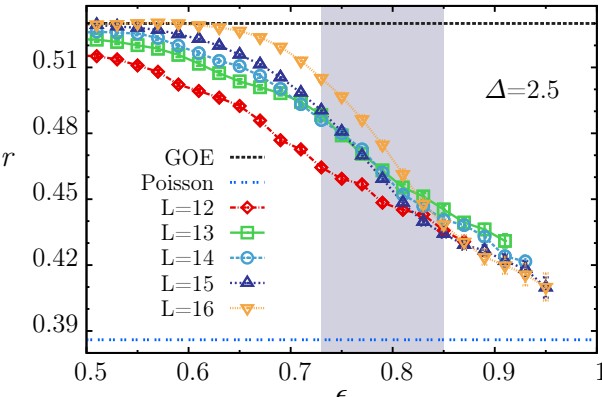

Figure 3: Adjacent-gap ratio $r$ as a function of the energy density $\epsilon$ for chains of variable size, and with $V = 2$ and $\Delta = 2.5$. The dashed and dotted lines correspond to the predictions for the Gaussian orthogonal ensemble (GOE) and Poisson level statistics, respectively. The grey-shaded area indicates the finite-size estimate for the location of the many-body mobility edge (see text).

## 3 MBL transition and many-body mobility edge

In this section we study the MBL transition in the full many-body spectrum of interacting spinless fermions in a quasi-periodic potential. The existence of a MBL regime for 1$d$ interacting

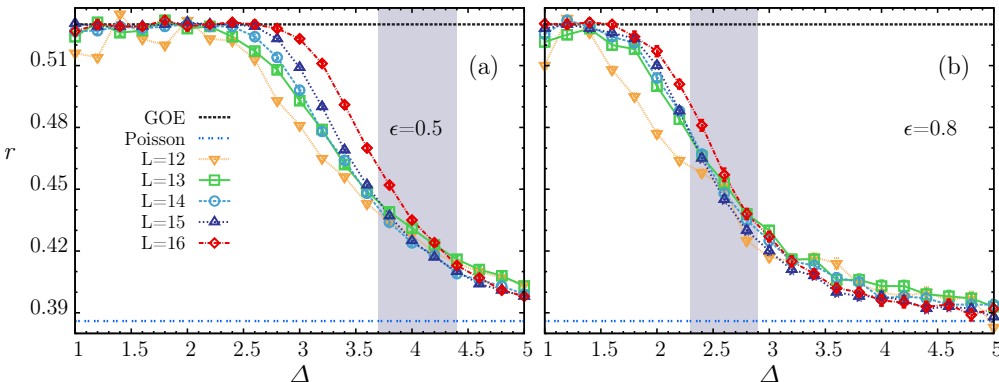

Figure 4: Adjacent-gap ratio $r$ as a function of the strength of the quasi-periodic potential $\Delta$ for two values of the energy density $\epsilon = 0.5$ and 0.8. Other parameters and symbols as in Fig. 3. The clear dependence of the transition region on the energy is a manifestation of the existence of a MBME.

spinless fermions in a strong quasi-periodic potentials has been already assessed in Ref. [32]. Here we move a step forward, reconstructing the MBL transition line as a function of the potential strength $\Delta$ and energy density. To do so, we choose for the interaction the value $V = 2$, which makes the fermionic chain equivalent to a $S = 1/2$ antiferromagnetic Heisenberg spin chain in a quasi-periodic magnetic field. As discussed in the previous section, studying the spectrum for a given value of $V$ is equivalent to studying the same spectrum with inverted energy axis for $-V$; in the case $V = -2$, the fermionic chain maps onto a $S = 1/2$ ferromagnetic Heisenberg chain. The ground-state localization transition for the repulsive fermionic (or antiferromagnetic spin) chain is estimated to occur at $\Delta_c = 2.5(1)$, whereas the transition for the attractive fermionic (ferromagnetic spin) chain is found to occur at $\Delta_c = 1.5(1)$. As mentioned in the previous section, one can expect the MBL transition line to connect continuously these two critical points.

## 3.1 Level statistics

In order to reconstruct the MBL transition line, we follow Refs. [6,20] and fully diagonalize the Hamiltonian $\mathcal{H}$ on small spin chains (up to $L = 16$) focusing on level statistics. To minimize finite-size effects we consider chains with periodic boundary conditions: while this introduces an artificial jump in the quasi-periodic potential on the bond connecting the $L-$th and the 1st site, this effect becomes negligible as the size increases [1]. Given the many-body spectrum $E_n$ of the Hamiltonian $\mathcal{H}$, one constructs the level spacings $\delta_n = E_n - E_{n-1}$ and defines the adjacent-gap ratio $r_n$ as the ratio between the minimum and maximum among two adjacent level spacings

$$r_n = \frac{\min(\delta_n, \delta_{n+1})}{\max(\delta_n, \delta_{n+1})} \ . \tag{3}$$

For each realization of the random phase $\phi$, the spectrum is normalized to its width

$$\epsilon_n = \frac{E_n - E_{\min}}{E_{\max} - E_{\min}} \tag{4}$$

---

[1]Using rational approximants for $\alpha$ in the form of $p/L$ for some integer $p$ eliminates the jump, but on the small system sizes we considered the resulting potential is very regular, often with a smaller period than the chain length. Therefore we refrain from using this approach.

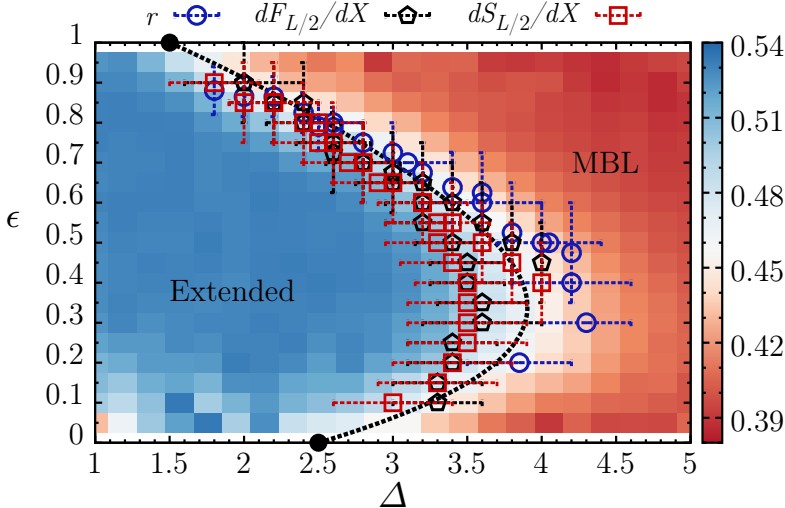

Figure 5: Spectral phase diagram of the repulsive fermionic chain with $V = 2$ in quasi-periodic potential, as a function of the potential strength $\Delta$ and of the energy density $\epsilon$. The phase boundary between the extended regime and the MBL regime is determined via the scaling of the $r$-ratio for between adjacent level spacings ($r$), the peak in the $X$-derivative ($X = \Delta, \epsilon$) of the entanglement entropy ($S$) and of the half-chain particle-number fluctuations ($F$) for a chain of length $L = 16$. The dashed line is a guide to the eye, and the false colors represent the $r$ ratio in the $L = 16$ chain.

and the ratios $r_n$ are then binned into groups corresponding to regular intervals of the energy density $\epsilon_n$ in the range $[0.05, 0.95]$. We discard the two tails at low and high energy in the spectrum on account of their extremely low density of states. More precisely, the $[0.05, 0.95]$ interval is divided into a grid of step $\Delta\epsilon$, and states are grouped into energy bins such that the $m$-th bin contains energy densities in the interval $[m \times \Delta\epsilon - \delta\epsilon, m \times \Delta\epsilon + \delta\epsilon]$. Here we have chosen $\Delta\epsilon = 0.02$ and $\delta\epsilon = 0.04$. The ratios $r_n$ are then bin-averaged, and further $\phi$-averaged over a large number of realizations, ranging from 1000 for $L = 16$ to 12000 for $L = 12$, in order to reconstruct the function $r_L(\epsilon; \Delta)$ for a given system size.

This function is shown in Fig. 3 for different chain lengths at fixed strength of the quasi-periodic potential $\Delta = 2.5$, focusing on the upper half of the spectral range, $\epsilon \in [0.5, 0.95]$. One clearly sees that, as the system size increases, the $r$ ratio close to the middle of the spectrum ($\epsilon = 0.5$) approaches from below the value $r_{\text{GOE}} = 0.5295(6)$ expected for energy levels which exhibit the level-spacing statistics of random matrices belonging to the Gaussian orthogonal ensemble (GOE) [6]. On the other hand, close to the upper spectral edge, $\epsilon \approx 1$, the $r$ ratio decreases with system size, approaching the value $r_{\text{Poisson}} = 2 \ln 2 - 1 \approx 0.386...$ which corresponds to level spacings being distributed according to Poissonian statistics. Our finite-size data (up to $L = 16$) do not quite reach the value $r_{\text{Poisson}}$ within the MBL region, but this is easily understood in that the we are working at fixed and intermediate strength of the quasi-periodic potential, which does not allow the localization length to become sizably smaller with respect to the system size. Despite the small system sizes considered here, we judge that the *inversion* in the scaling of $r$ upon increasing $\epsilon$ – exhibited by the data in Fig. 3 – offers a rather strong indication of the existence of a many-body mobility edge (MBME). In particular the scaling inversion suggests that, in the thermodynamic limit, the $r$ ratio will jump in between the two above values at a MBME separating an extended phase obeying the GOE statistics of level spacings and exhibiting level repulsion; and an MBL phase obeying the Poisson statistics of level spacings, namely lacking level repulsion. A finite-size estimate for

the location of the MBME can be obtained via the crossing of the $r_L$ curves for different system sizes. In particular the transition "region" (shaded in grey in Fig. 3) is estimated as the region containing the crossing points between the four largest sizes considered ($L = 14$ with $L = 16$, and $L = 13$ with $L = 15$ [2]), and marking therefore the scaling inversion for these sizes. The width of the so-estimated transition region constitutes the total error-bar width attached to the transition points shown in Fig. 5.

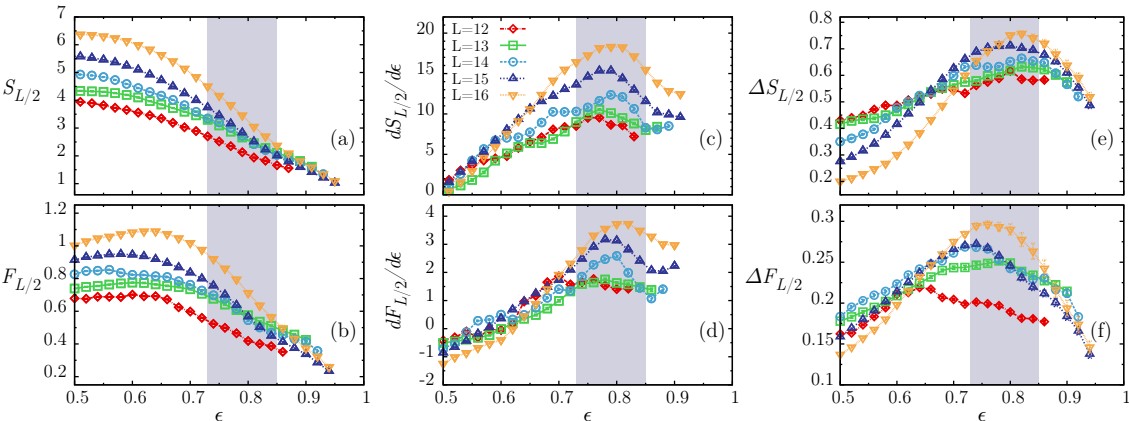

Figure 6: Entanglement entropy and particle-number fluctuations across the many-body mobility edge. All data refer to the model with $V = 2$, $\Delta = 2.5$ and with different lattice sizes. (a) Entanglement entropy; (b) particle-number fluctuations; (c) $dS_{L/2}/d\epsilon$; (d) $dF_{L/2}/d\epsilon$; (e) $\Delta S_{L/2}$ and (f) $\Delta F_{L/2}$. The shaded area corresponds to the estimate of the mobility edge from the scaling of the adjacent-gap ratio, Fig. 3.

This same analysis is repeated for different values of $\Delta$, reconstructing in this way the $\Delta$-dependence of the MBME. Moreover we also estimate the MBME by monitoring the evolution of the level statistics at fixed energy and variable $\Delta$, as shown in Fig. 4. Our findings are collected in Fig. 5, showing that the locus of MBMEs seemingly interpolates between the $\epsilon = 0$ and the $\epsilon = 1$ transitions, yet in a very non-trivial manner, namely with a belly-shaped curve featuring a wide re-entrance of the extended phase as one moves away from the two edges of the spectrum. This qualitative feature is also observed in the estimated mobility edge for the widely investigated case of the Heisenberg spin chain in a random field [20–22, 24, 27, 43] as well as in other disordered models [19, 23]. It is a consequence of the fact that, starting from a localized ground state (or most excited state) at $\epsilon = 0$ ($\epsilon = 1$), and increasing (decreasing) the energy density, localization gets weaker. This is generically understood from the limit of a classical lattice gas (namely $j = 0$), which features an increasing density of states upon moving away from the edges of the spectrum; reintroducing the hopping in the system, the latter can couple coherently an increasing number of quasi-resonant states, leading *e.g.* to a growing entanglement entropy of the half system. This argument implies that the maximum critical disorder strength to induce MBL is expected for $\epsilon \approx 0.5$, featuring the maximum density of states for all potential strengths. This latter feature implies therefore the belly-shaped MBME line seen in Fig. 5.

---

[2]We use the crossing between system sizes of the same parity because of even-odd effects which are observed on our small samples. These are due primarily to the fact that the half-filling condition imposed to our calculation is satisfied differently for even and odd sizes.

## 3.2   Scaling of entanglement entropy and fluctuations

The estimate of the MBL transition obtained from the statistics of level spacings is corroborated by an investigation of the macroscopic properties of the states in the spectrum, namely the entanglement and particle-number fluctuations of the half chain. We evaluate the half-chain entanglement entropy of Eq. (2) $S_{L/2,n}$ on each excited state $|\psi_n\rangle$, as well as the particle-number fluctuations on the half chain

$$F_{L/2,n} = \langle\psi_n|N_{L/2}^2|\psi_n\rangle - \langle\psi_n|N_{L/2}|\psi_n\rangle^2 \tag{5}$$

where $N_{L/2} = \sum_{i=1}^{L/2} n_i$ ($L/2 \to (L-1)/2$ for odd-sized chains). Similarly to the adjacent gap ratio, the entropy $S_{L/2,n}$ and fluctuations $F_{L/2,n}$ of individual eigenstates are bin-averaged corresponding to their energy density $\epsilon_n$, and subsequently the bin averages are further averaged over the phase $\phi$ to reconstruct the functions $S_{L/2}(\epsilon;\Delta)$ and $F_{L/2}(\epsilon;\Delta)$. The latter functions are shown in Fig. 6 for the same cut in the $(\Delta,\epsilon)$ plane as that of Fig. 3.

One remarkably observes that, in correspondence with the estimate of the MBME from the scaling of the adjacent-gap ratio, the half-chain entanglement entropy and particle-number fluctuations change in their scaling behavior: they appear to grow like the system size in the putative extended phase (revealing a volume law), and not scale at all in the MBL phase (revealing an area law). Hence in the thermodynamic limit the entropy and fluctuations should exhibit a violent decrease at the MBL transition upon increasing the energy, resulting in a maximum derivative (and possibly a divergent one) with respect to $\epsilon$ at the transition point. This feature survives also in finite-size systems, exhibiting a clear maximum in the energy derivative of the entanglement entropy and fluctuations (see Fig. 6(c-d)): hence the position of the maximum can be taken as a finite-size estimate of the MBL transition. A similar reasoning can be used for the dependence of the entanglement entropy and fluctuations on $\Delta$: a maximum in the $\Delta$-derivative can be used again as a finite-size estimate of the MBL transitions. These estimates (from the $\epsilon-$ and $\Delta-$ derivatives) for a system size of $L = 16$ are reported in Fig. 5, and they allow to draw a phase boundary between the extended and MBL phase which is in good agreement with the MBME obtained from the scaling of the adjacent-gap ratio.

Finally, we analyze the adjacent-state variation of the entanglement entropy and fluctuations

$$\begin{aligned} \Delta S_{L/2,n} &= |S_{L/2,n} - S_{L/2,n-1}| \\ \Delta F_{L/2,n} &= |F_{L/2,n} - F_{L/2,n-1}| \end{aligned} \tag{6}$$

as shown in Fig. 6(e-f) – where we report the bin- and phase-averaged quantities $\Delta S_{L/2}$ and $\Delta F_{L/2}$. According to ETH, the entropy and fluctuations are smooth functions of the energy of the eigenvalues, and therefore the above these differences should be exponentially small in the system size. Indeed the energy difference between two adjacent eigenstates is of the order $\mathcal{O}[L\exp(-L)]$, and therefore a similar scaling should also be exhibited by the entropy and fluctuation differences, if one requires differentiability of the entropy and fluctuations with respect to the energy. A decreasing behavior with system size in both $\Delta S_{L/2}$ and $\Delta F_{L/2}$ is indeed observed in the putative extended phase, changing drastically into an increasing behavior upon approaching the transition to the localized regime. In particular the two above quantities exhibit a peak roughly in correspondence with the estimated mobility edge: this can be justified by observing that the entropy and fluctuations go from volume-law scaling to area-law scaling upon crossing the transition, and therefore the corresponding state-to-state variations (which at most obey the same scaling) exhibit their largest magnitude close to the transition.

In conclusion, we argue that our results from exact diagonalization on small sizes already provide strong evidence for the existence of a mobility edge in the many-body spectrum. In

particular for $\Delta = 2.5$ and $V = 2$ we estimate the position of the MBME as $\epsilon = 0.79(6)$. As we shall see in the next section, an even stronger evidence in favor of the MBME is offered by the out-of-equilibrium dynamics.

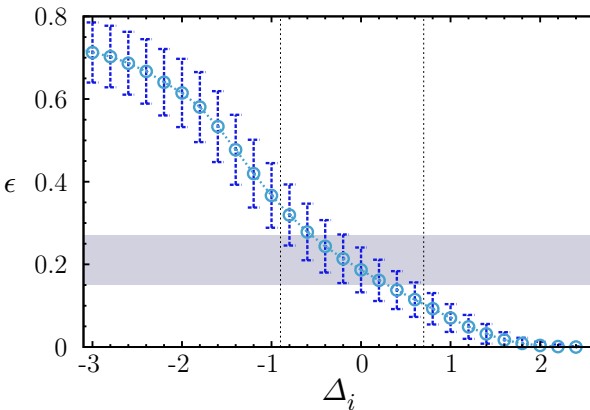

Figure 7: Energy density injected into the system after a quench from an initial potential depth $\Delta_i$. The data refer to a system with $\Delta_f = 2.5$, $V = -2$, and $L = 16$. The error bars represent the standard deviation of disorder fluctuations, $(\delta \epsilon)_{\text{dis}}$. The grey area indicates again the estimate of the mobility edge from the level statics.

## 4 Quench dynamics and quench spectroscopy

The disorder-driven MBL transition at fixed energy, or the energy-driven MBL transition at fixed disorder across a MBME, represent most intriguing features of many-body localization, as well as possibly the most challenging ones for a quantitative study. Indeed while MBL states have a low (area-law) entanglement content, and therefore admit an efficient representation in one dimension as matrix-product states [44], extended states do not share the same property. This aspect restricts significantly the simulability of the MBL transition, and essentially limits its study to exact diagonalization. In this respect, experimental quantum simulation of closed quantum systems, such as that offered by ultracold atoms in controlled optical potentials [45], appears as a viable route to test this transition beyond the capabilities of numerical approaches, as already explored in recent seminal studies [12–15]. So far the experimental investigations have essentially focused on the disorder-driven transition as detected by preparing the system in a highly excited region of the spectrum, but they otherwise lack energy resolution. A similar consideration can be made for most of the theoretical studies of MBL based on the analysis of the quasi-stationary state after a quantum quench [17,26,32,46–48], namely on the long-time properties of the evolved state

$$|\psi(t)\rangle = e^{-i\mathscr{H}t}|\psi(t=0)\rangle \tag{7}$$

starting from a high-energy initial state $|\psi(t=0)\rangle$. The initial state is typically chosen as a factorized state (either random or periodic), with energy density $\epsilon \approx 0.5$.

### 4.1 Principle of quench spectroscopy

Our present work has the ambition of proposing a protocol for the experimental and theoretical study of MBL, which possesses energy resolution, and which therefore is ideally suited to the

detection of MBMEs. Our protocol – that we dub *quantum-quench spectroscopy*, and which is sketched in Fig. 1 – is based on the injection of a controlled amount of energy density in the system, described by the (final) Hamitonian $\mathscr{H}_f$. The energy control is achieved by the choice of the initial state $|\psi(t=0)\rangle$ as ground state of an initial Hamiltonian $\mathscr{H}_i$ which is connected to $\mathscr{H}_f$ by a continuous parameter change. For the system under examination, a natural choice of the parameter in question is the strength $\Delta$ of the quasi-periodic potential, which can be continuously controlled by the intensity of a laser potential in cold-atom setups [12,33]. Indicating with $|\psi_i^{(0)}\rangle$ and $|\psi_f^{(0)}\rangle$ the ground states of the initial and final Hamiltonian respectively, a sudden quench $\Delta_i \to \Delta_f$ transfers to the system the energy

$$\Delta E = \langle \psi_i^{(0)}|\mathscr{H}_f|\psi_i^{(0)}\rangle - \langle \psi_f^{(0)}|\mathscr{H}_f|\psi_f^{(0)}\rangle \tag{8}$$

or, equivalently, an energy density $\epsilon = \Delta E / W$, where $W$ is the spectral width of the final Hamiltonian. Obviously the energy-density transfer $\epsilon$ comes with an intrinsic quantum uncertainty

$$(\delta\epsilon)_{\text{int}} = \frac{1}{W}\sqrt{\langle \psi_i^{(0)}|\mathscr{H}_f^2|\psi_i^{(0)}\rangle - \langle \psi_i^{(0)}|\mathscr{H}_f|\psi_i^{(0)}\rangle^2} \tag{9}$$

which nonetheless scales to zero as $L^{-1/2}$ when the system size increases, due to the fact that both the spectral width $W$ and the quantum uncertainty on the $\mathscr{H}_f$ operator (or the expression under the square root in the previous equation) grow linearly with system size. As a consequence, the transferred energy density $\epsilon$ is a sharply defined quantity in sufficiently large systems for a given realization of the (quasi-)disordered potential. A further source of uncertainty on the injected energy density comes from the fluctuations of the injected energy density between different disorder realizations, estimated as the standard deviation $(\delta\epsilon)_{\text{dis}}$ of $\epsilon_\phi$ injected for a random phase $\phi$. In the system sizes under investigation ($L = 14$ and 16), we find that $(\delta\epsilon)_{\text{dis}} \gtrsim (\delta\epsilon)_{\text{int}}$, and later we use $(\delta\epsilon)_{\text{dis}}$ as the uncertainty attached to the $\epsilon$ injected in a quench. The disorder fluctuations $(\delta\epsilon)_{\text{dis}}$ of the energy density can also be expected to decrease with increasing system size, due to the self-averaging nature of the $\mathscr{H}_f$ operator, although possibly more slowly than $(\delta\epsilon)_{\text{int}}$, due to the correlated nature of the quasi-periodic potential.

The injected energy density $\epsilon$ into can be continuously controlled by fixing $\Delta_f$, and continuously varying $\Delta_i$. To investigate the dynamical signatures of a MBME, we choose $\Delta_f = 2.5$ and $V = -2$, namely we shall hereafter focus on *attractive* spinless fermions. As discussed in Sec. 2, this amounts to reading the phase diagram of Fig. 5 with an inverted $\epsilon$ axis, namely sending $\epsilon$ into $1-\epsilon$. Based on the analysis of Sec. 3, this Hamiltonian has a unique MBME for an energy density $\epsilon \approx 0.21$ above its ground states (corresponding to a MBME at $\epsilon \approx 0.79$ for the repulsive model with $V = 2$). Fig. 7 shows the energy density injected into the attractive model by a parametrically varying quantum quench as a function of $\Delta_i < \Delta_f$, resulting from an exact-diagonalization calculation for $L = 16$ averaged over different values of the phase $\phi$. One can observe that this quench scheme allows to inject an essentially arbitrary amount of energy density into the system, and particularly so when the quasi-periodic potential changes sign, so that, after the quench, particles trapped around the minima of the initial potential find themselves sitting over the maxima of the final potential. A critical value $\Delta_{i,c} \approx 0$ appears to separate small quenches with $\Delta_i > \Delta_{i,c}$, which shall leave the system in the MBL phase, from large quenches $\Delta_i < \Delta_{i,c}$, which shall bring the system across the MBME into the extended phase.

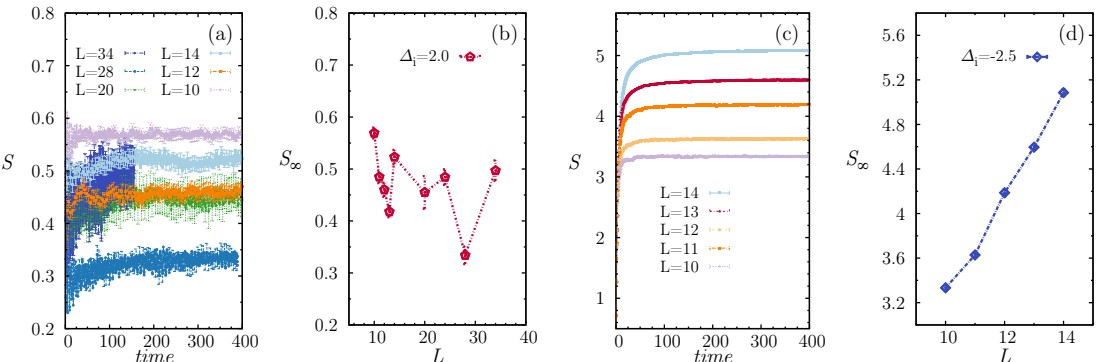

Figure 8: Entanglement entropy after a quantum quench. (a) Evolution after a quench from the ground state with $\Delta_i = 2$, as obtained from exact diagonalization and t-DMRG. The results are averaged over more than 400 disorder realizations for $L \leq 14$ and for 120 realizations for the larger sizes. (b) Scaling of the average over the final interval, $S_\infty$ (see text); (c) Evolution after a quench from the ground state with $\Delta_i = -2.5$, obtained from exact diagonalization; (d) scaling of the average over the final interval, $S_\infty$.

## 4.2 Long-time dynamics: extended vs. localized quasi-stationary states

Focusing on the attractive model with $V = -2$ and $\Delta_f = 2.5$, we investigate the long-time dynamics and the quasi-stationary state of the evolution making use of exact diagonalization on lattices up to $L = 14$, and time-dependent DMRG (t-DMRG) [49–51] on lattices up to $L = 34$ Our t-DMRG simulations are based on the Runge-Kutta scheme, and used up to $M = 500$ states allowing us to keep the truncation error smaller than $10^{-7}$ at each time step. Hereafter we will set $\hbar = 1$ and measure the time in units of the hopping parameter $j$. All simulations have been run till a time $t_{\max} = 400$, except those for $L = 34$, which have been stopped at $t_{\max} = 150$.

Fig. 8(a) shows the entanglement entropy after two quenches of widely different width. In the case of the small quench ($\Delta_i = 2$) – leading to a weak energy-density injection – one clearly observes a very weak growth of the half-chain entanglement entropy. As a result, the entanglement entropy in the quasi-stationary state $S_\infty$ - obtained as the time average over the final time interval $\Delta t = [300, 400]$ ([100, 150] for $L = 34$) shows essentially an absence of scaling, as detailed in Fig. 8 (b). This is a clear signature of a complete lack of thermalization, characteristic of MBL, whose benign side effect is a very weak entanglement content of the evolved state, allowing to scale up the simulation by the use of t-DMRG. These results are a very solid manifestation of the existence of a low-energy MBL regime, corresponding to the analysis of the previous section.

On the other hand, a large quench ($\Delta_i = 2.5$) injects sufficient energy to change completely the behavior of the asymptotic state. The half-chain entanglement entropy converges to a value $S_\infty$ which clearly scales linearly with the system size – as shown in Fig. 8 (d). The fast growth of entanglement with system size in turn limits the study of large quenches to exact diagonalization. The linear scaling of entanglement entropy suggests that the quasi-stationary state may correspond to a thermalized state, meaning that large quenches give access to the extended states of the Hamiltonian sitting beyond the MBME.

To make the latter statement fully quantitative, we have analyzed the quasi-stationary *entanglement entropy density*, defined as the linear coefficient $s_Q$ of the volume-law scaling term of the asymptotic entanglement entropy $S_\infty = s_Q L/2 + c$ (where $c$ is a constant). For the quasi-stationary state to be extended according to ETH, the entanglement entropy density $s_Q$ should correspond to the microcanonical entropy density $s_m(\epsilon)$. The latter can be estimated

via exact diagonalization, by extracting for each lattice size $L$ the microcanonical entropy

$$S(\epsilon; L) = \log \Omega(\epsilon; L) \tag{10}$$

where $\Omega(\epsilon; L)$ is the number of states in a narrow energy window around the energy density $\epsilon$ [3]. A fit to the form $S(\epsilon; L) = s_m(\epsilon)L + c'$ (with $c'$ a constant) allows to extract the entropy density.

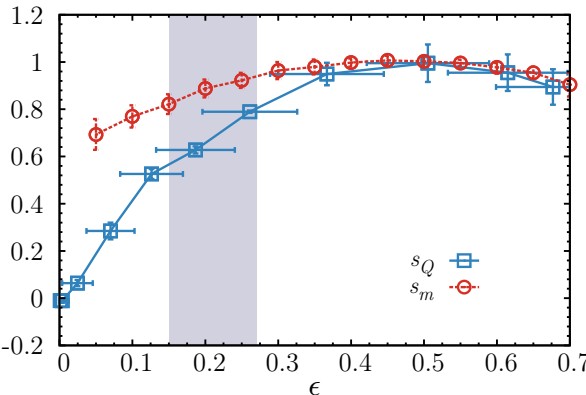

Figure 9: Entanglement entropy density $s_Q$ from the scaling of the quasi-stationary entropy $S_\infty$ after a quench as a function of the injected energy density, compared with the microcanonical entropy density $s_m$. The shaded area marks the estimate for the location of the mobility edge.

A direct comparison between $s_Q(\Delta_i)$ and the microcanonical entropy density $s_m(\epsilon)$ is possible when using the correspondence established in Fig. 7 between the initial depth of the quasi-periodic potential $\Delta_i$ before the quench, and the energy density injected into the system after the quench. This comparison is carried out in Fig. 9 and it provides a very interesting insight. Indeed one sees that for small energy densities injected by the quench, the entanglement entropy density and the thermodynamic entropy are radically different: namely, even if the post-quench quasi-stationary state displays an extensive entanglement entropy (unlike each of the eigenstates in the MBL regime), it remains very far from the thermal state. On the other hand, for a sufficiently large energy density the entanglement entropy density matches almost perfectly the thermal value. This is a further, strong evidence – coming from the out-of-equilibrium evolution – that the system with $\Delta = 2.5$ possesses a spectral range with extended states. This indication, along with the strong evidence of MBL for sizes up to $L = 34$ for small quenches, corroborates again the existence of a MBME in the spectrum.

The estimate of the position of the MBME from the value of $\epsilon$ at which $s_Q$ matches $s_m$ turns out to be in reasonable agreement with the estimate extracted from the scaling analysis on the properties of the spectrum reported in Sec. 3. Our finite-size limitations may easily lead to an underestimation of $s_Q$ with respect to its true value in the thermodynamic limit; a similar reasoning can be made for finite-time effects due to possible incomplete saturation of the entanglement entropy to its asymptotic long-time limit. Despite the uncertainty in the location of the MBME from the quench dynamics, it remains nonetheless incontrovertible that two distinct regimes (an extended one and an MBL one) exist in the properties of the quasi-stationary state. As we shall see in the next section, the short-dynamics provides an even richer palette of dynamical behaviors, with a seemingly universal time dependence in the MBL regime, and a disorder- and energy-dependent behavior in the extended one.

---

[3]As usual the particular width $w$ of the energy window does not matter because the entropy depends on it as $\log(w)$, namely this dependence becomes irrelevant in the thermodynamic limit.

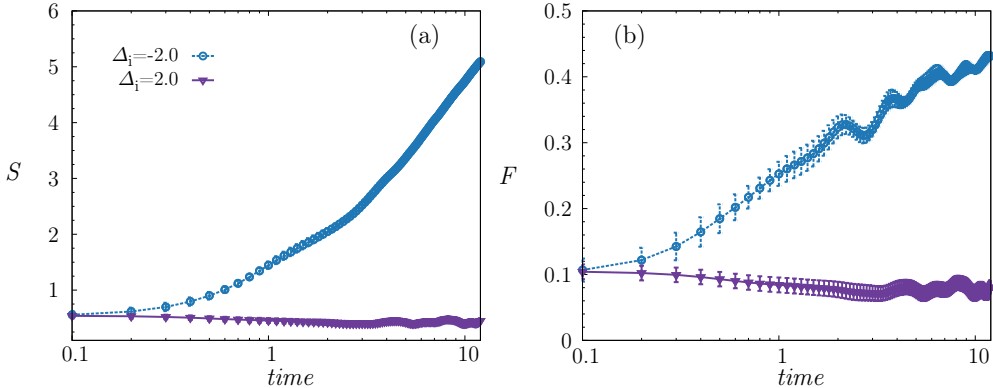

Figure 10: Short-time evolution of (a) the entanglement entropy and (b) the fluctuations for two quenches from the ground state with $\Delta_i = \pm 2$.

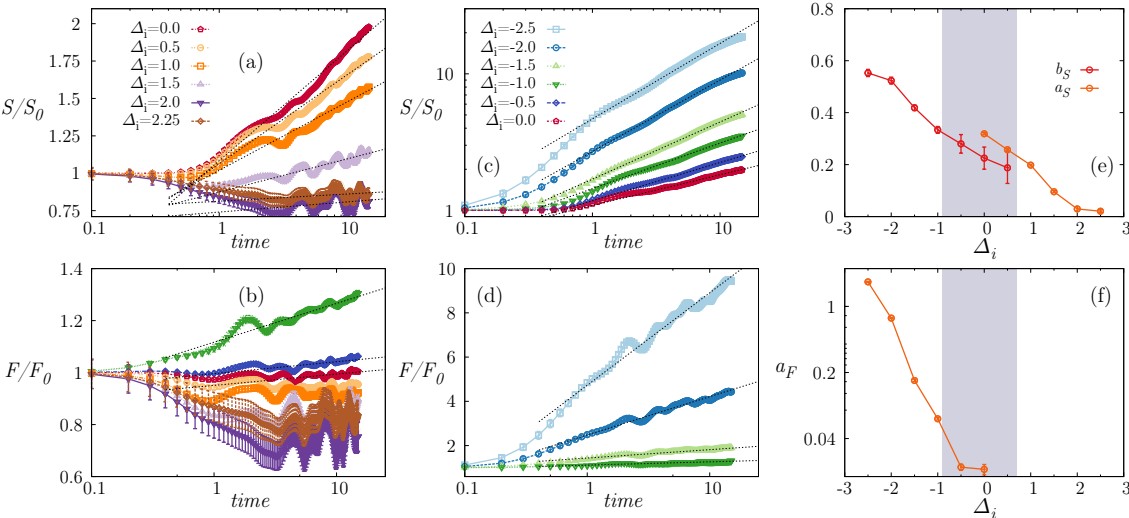

Figure 11: Short-time evolution after quenches of variable width. The results are averaged over more than 150 disorder realizations on a lattice of L = 18. (a) Entanglement entropy and (b) fluctuations after a quench from the ground state with $\Delta_i \geq 0.0$, leaving the evolved state in the MBL regime. The dashed lines are fits to $S(t) = a_S \log t + \text{const}$ for the entanglement entropy. To improve readability, the data are renormalized to the values $S_0$ and $F_0$ at time $t = 0$. (c) Entanglement entropy and (d) fluctuations after a quench from the ground state with $\Delta_i \leq 0$, preparing the system into a superposition of extended states. The dashed lines are fits to $S(t) = b_S t^{\alpha} + \text{const}$ for the entropy, and to $F(t) = a_F \log t + \text{const}$ for the fluctuations. Fit coefficients (e) $a_S$ and $\alpha$ and (f) $a_F$ as a function of $\Delta_i$. The shaded area indicates the critical value of $\Delta_i$ required to prepare the system in a superposition of extended states.

## 4.3 Short-time dynamics: dynamical transition

The stationary-state properties after quantum quenches of variable width provide invaluable information on the nature of the Hamiltonian eigenstates which are "populated" by the quench itself. Nonetheless reaching the asymptotic quasi-stationary behavior is challenging both numerically (on sufficiently large system sizes) as well as experimentally (due to decoherence and

dissipation effects). At the same time, the transient dynamics of relaxation towards the quasi-stationary state of strongly disordered interacting systems turns out to contain even richer information, concerning the properties of particle transport and information transport in the system.

The MBL regime was numerically shown [16, 17] and later predicted [52, 53] to exhibit a logarithmically slow growth of entanglement entropy of a subsystem. This behavior is understood as the result of exponentially small (in the distance) effective couplings existing between localized degrees of freedom (or so-called "local integrals of motion") [52, 54, 55] whose excitations, activated by the quench, are nearly resonant in energy, and therefore can be communicated over the small couplings to induce entanglement. On the other hand, after a short transient particle-number fluctuations were numerically shown not to grow with time in the MBL phase [26]. In the extended regime close to the MBL transition, on the other hand, a slow dynamics has been predicted and numerically verified [29–31, 48] to occur, characterized by dynamical exponents which are continuous functions of disorder, and which vanish at the onset of MBL.

Hence the MBL transition is characterized by a continuous transition in the relaxational dynamics, which has been recently verified numerically as a function of disorder at fixed energy density [31]. Since our main focus is the study of the energy-driven MBL transition across a MBME, we shall search for a similar phenomenology in the dynamics after variable quantum quenches, preparing the system below or above the MBME in the system. A first indication of the extremely strong energy dependence of the post-quench dynamics is provided in Fig. 10, which shows the evolution of the entanglement entropy and fluctuations after two quenches with $\Delta_i = 2$ and $-2$, corresponding to a small and large quench amplitude respectively. Interestingly, the initial state has the exact same properties in the two cases – given that, on average over the phase $\phi$, the equilibrium properties of the system are invariant under a change in sign of the quasi-periodic potential. Yet the ensuing dynamics after a quench to $\Delta_f = 2.5$ is radically different: the small-quench dynamics leads to a decrease in the entanglement entropy and fluctuations, witnessing the weak entanglement and fluctuation content of the weakly excited states of the final Hamiltonian; whereas both entanglement and fluctuations substantially increase after the large quench.

A thorough analysis of the dynamical transition bridging the above regimes of small and large quenches is provided by Fig. 11. There we observe that, after an initial transient, *small quenches* (namely $\Delta_i \gtrsim 0$ or $\epsilon \lesssim 0.2$) lead to 1) a logarithmic increase in the entanglement entropy, $S(t) \sim a_S(\Delta_i) \log t$ [Fig. 11(a)]; with $a_S(\Delta_i)$ depending continuously on the amplitude of the quench [Fig. 11(e)]; 2) no visible increase in the fluctuations [Fig. 11(b)]; both features are compatible with those previously observed for the MBL regime. This again confirms that the low-energy sector of the Hamiltonian spectrum is dominated by localized states.

On the other hand, *large quenches* (namely $\Delta_i \lesssim 0$ or $\epsilon \gtrsim 0.2$) lead to a radically different dynamics: 1) the entanglement entropy exhibits an $\epsilon$- (or $\Delta_i$-)dependent slow dynamics [Fig. 11(c)], $S(t) \sim t^\alpha$ with $\alpha = \alpha(\Delta_i) \to 0$ when $\Delta_i$ approaches the value necessary to prepare the state with an energy beyond the MBME [Fig. 11(e)]; 2) the fluctuations exhibit a logarithmically slow increase with time [Fig. 11(d)], $F(t) \sim a_F(\Delta_i) \log t$, with a coefficient $a_F(\Delta_i)$ which seemingly vanishes upon approaching the MBME [Fig. 11(f)]. The evolution of entanglement entropy is fully compatible with the behavior of an extended regime lying close to the MBL transition found in Refs. [29–31], whereas the logarithmic growth of fluctuations has been observed very recently in a similar regime in Ref. [56]. The radical difference between the evolution of entanglement and that of fluctuations, while *per se* very intriguing, can be understood by observing that the particle number is constrained by a conservation law, while entanglement can be generated freely [57]. Refs. [29, 30] predict that, in the extended phase close to the MBL, the energy transport is subdiffusive, namely the characteristic change

$\delta E$ in the local energy evolves as $\delta E \sim t^{\beta}$ with a dynamical exponent $\beta$ which is related to that of the entanglement as $\beta = \alpha/(1 + \alpha)$. Since energy transport can potentially occur only via collisions and without particle transport, one can imagine that $F \sim t^{\beta'}$ with $\beta' < \beta$ (and $\beta' \approx 0$ from our numerical results). Therefore the extended phase close to the MBL transition may be as rich as to feature at least three distinct forms of dynamics (associated with particle transport, energy transport and entanglement spreading, in increasing order of speed).

## 5    Conclusions

In this paper we have proposed and numerically demonstrated a technique (quantum-quench spectroscopy) to probe the localized vs. extended nature of selected regions in the spectrum of a strongly disordered, interacting quantum systems. Quantum-quench spectroscopy provides in particular a probe for the existence and the dynamical signatures of a many-body mobility edge. Preparing the system in an initial state, which is chosen to be parametrically far from the ground state of a given target Hamiltonian, allows to selectively probe different energy sectors of the spectrum of the Hamiltonian in question. We have applied quantum-quench spectroscopy to a model of one-dimensional interacting fermions in a quasi-periodic potential, possessing strong numerical signatures for the existence of a many-body mobility edge in the spectrum over a broad range of potential strengths. When the low-energy sector of the Hamiltonian is many-body localized, the quasi-stationary state after a small quench deviates radically from a thermal state, and the eigenstate thermalization hypothesis (ETH) is not verified. On the other hand, when the quench is sufficiently large as to inject an energy exceeding the many-body mobility edge, the quasi-stationary state after long-time dynamics is found to become ergodic. In particular, its entanglement entropy matches the microcanonical entropy at the corresponding energy, as mandated by ETH. Even more strikingly, the relaxation dynamics towards the quasi-stationary state is radically different in the two regimes of small and large quenches: below the many-body mobility edge the entanglement entropy grows logarithmically with time, and particle number fluctuations do not show any growth; above the many-body mobility edge, the entanglement entropy grows with time as a power law, with a dynamical exponent parametrically depending on the energy, whereas the particle number fluctuations appear to increase logarithmically.

Our results have a deep significance both at the theoretical as well as at the experimental level. On the theory side, we provide ample evidence for the existence of a many-body mobility edge in the model under investigation. Despite the smallness of the lattice sizes we can treat numerically with exact diagonalization, the scaling of the spectral properties clearly points towards this conclusion. Moreover, quantum-quench spectroscopy serves as a new theoretical diagnostics giving invaluable insight already at the level of the short-time dynamics. The latter can be studied with time-dependent DMRG, scaling up significantly the system sizes that are numerically accessible. The remarkable agreement between the analysis of the spectral properties and the analysis of the quench dynamics point towards the existence of a transition in the spectrum.

At a more general level we observe that, in the recent literature, quench dynamics is typically used to exhibit the different dynamics driven by *different* Hamiltonians, starting from the *same* initial state, or from a random state generated from the same distribution – this is the case *e.g.* for seminal studies of many-body localization both with numerics [16, 17, 32] and with cold-atom experiments [12, 15]). Here we show that strongly disordered quantum systems offer a new paradigm, in which the *same* Hamiltonian can lead to radically different dynamics when initializing the system in *different* initial states, due to the existence of a transition in the spectrum.

On the experimental side, quantum-quench spectroscopy lends itself very naturally to an implementation within cold-atom setups, endowing the quantum simulation of many-body localized systems with energy resolution. In particular cold-atom setups offer an invaluable, microscopic insight into the dynamics of the system via the use of quantum-gas microscopes [58], already applied to many-body localized systems in a very recent experiment [14]. Microscopy provides potential insight into transport properties and into the spreading of correlations [59] which, based on our results, are expected to possess radically different features in the extended vs. localized regimes already at the level of the (experimentally accessible) short-time dynamics. The study of the signature of the MBME transition in the short-time spreading of correlations after the quench shall be the subject of future studies.

# 6 Acknowledgements

We thank C. Degli Esposti Boschi, F. Ortolani, L. Taddia, and D. Vodola for useful discussions. T.R. acknowledges the support of ANR ("ArtiQ" project). P.N. and E.E. acknowledge financial support from the INFN grant QUANTUM. The QMC calculations where performed on the HPC facilities of PSMN (ENS de Lyon).

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
