# Peer review of "Detecting a many-body mobility edge with quantum quenches"

_SciPost Physics, doi:SciPost Phys. 1, 010 (2016)_

## Round 1 · Referee Report · Anonymous (Referee 1) · 2016-9-8

Strengths

1- Very thorough analysis of ground-state , fintie energy density and out-equilibrium properties of the model.

2- Broad spectrum of theoretical methods employed in the work (exact diagonalization, DMRG, QMC)

3- Original proposal for the observation of the many-body mobility edge

4- Well written manuscript, very good presentation

Weaknesses

1- More concrete predictions for experimentally accessible observalbles besides particle number fluctuations would be desirable.

2- Some few omissions of related literature

Report

Report

The prediction of the existence of a many-body mobility edge in
interacting systems with quenched disorder is one of the key
results of the theory of many-body localization. So far there,
are no experimental attempts at verifying the existence of the
many-body mobility edge. Moreover, the very existence of a many-body
mobility edge has been questioned by some theoretical papers.

The present manuscript makes a concrete proposal for the observation
of the many-body mobility edge using a quench protocol that involves
quenching the strength of a quasi-periodic potential, which are used
in many optical lattice experiments to study localization physics.

The paper contains a very thorough analysis of the equilibrium properties
of the model, both in the ground state as well as for finite energy densities.
Most notably, this analysis is summarized in ground-state and high energy
phase diagrams, the latter suggesting the existence of a many-body mobility edge in the model.

The authors proceed to study entanglement dynamics to characterize the quantum
quench dynamics as a measure for whether the post-quench state is in the
delocalized or localized part of the many-body spectrum. The results suggest that the many-body mobility edge could be resolved from both the long-time limit
and the short time-dynamics of entanglement measures.

The paper is very timely, of clear relevance to the field, very well written
and the results and conclusions are convincing. The results will be of interest
to both theoretical and experimental researchers interested in MBL. With regard
to experiments, one must admit that measuring entanglement in a disordered system is not possible yet. Thus the most relevant predictions for an experiment are those for the particle number fluctuations. Perhaps the authors could elaborate more on what an experimentalist should measure in this particular quench set-up.

Other than that, I have only very minor suggestions and comments (see below)
and I recommend publications of the manuscript after the authors have given
considerations to the points mentioned here.

Requested changes

1- If possible, then it would be desirable to elaborate more on the behavior of experimentally accessible observables (other than particle number fluctuations) and signatures of the mobility edge in their dynamics.

2- The observation of an area law in the MBL goes back to a paper by Bauer and Nayak, which in my opinion should be cited here.

3- TDMRG was developed in parallel efforts by White and Feiguin and Daley et al. J. Stat. Mech.: Theory Exp. (2004), P04005. The latter reference is missing, perhaps it would also be fair to cite Vidal Phys. Rev. Lett. 93, 040502 (2004) as well, which was very important in the methodological developments.

4- Figure 2 shows little contrast between the LL and the CDW phase. Isn't that just a coincidence? in the CDW phase, there should be an area law and hence for large L, the half-cut entanglement in the LL phase should exceed that in the CDW phase.

5- By comparing ground state and finite energy density properties, can the authors speculate about the existence of an inverted mobility edgde (i.e., a transition from the LL into the MBL phase as energy density increases)?

6- By inspection of Fig. 8 it seems that the estimate of the position of the many-body mobility edge from the quantum quench disagrees with the result of equilibrium calculations by a factor of 2. Perhaps this could simply be states, the authors on page 8, just before Sec C is perhaps a bit too optimistic.

7- There is no final agreement yet about the existence of a sub-diffusive regime in the ergodic phase. Numerical results for sigma_dc consistently report sigma_dc >0, inconsistent with subdiffusive dynamics (see Steinigeweg et al. arXiv:1512.08519 and Barisic et al. Phys. Rev. B 94, 045126 (2016)).

8- I'd like to draw the authors' attention to Phys. Rev. Lett. 115, 230401 as well, another study of MBL in a 1D system with a quasi-periodic potential.

  • validity: top
  • significance: high
  • originality: high
  • clarity: top
  • formatting: excellent
  • grammar: excellent

Author:  Tommaso Roscilde  on 2016-10-21  [id 70]

(in reply to Report 1 on 2016-09-08)
Category:
answer to question
reply to objection

We would like thank the Referee for the appreciation expressed towards our work, and for the useful suggestions. We reply to the requests one by one:
1 - This is a delicate point, and, as a matter of fact, it is the subject of our current investigations. As we already stated in the previous version of the manuscript, we believe that quantum-gas microscopy can extract salient features of the quench-induced MBL transition by monitoring the dynamics of correlation spreading, as done in famous previous experiments on non-disordered one-dimensional lattice gases. The precise form of the dynamics of correlations is a subject on its own, which in our opinion goes well beyond the scopes of the present manuscript. Nonetheless we have expanded our discussion of this experimental signature in the coda of the manuscript;
2 - We agree with the suggestion, and we have added the corresponding citation;
3 - Same as at point 2;
4 - The little contrast is a coincidence, as discussed already in the text: the slow logarithmic size dependence of entanglement in the LL phase produces a half-chain entropy which, for the system sizes we considered, is nearly equal to that produced by the double degeneracy of the ground state in the CDW phase;
5 - We could not observe such an inverted mobility edge in our calculations -- in the model under investigation a ground-state LL phase always evolves into an extended phase as the energy density is turned on. It would obviously be very fascinating to observe such a counterintuitive phenomenon, although we are not aware of a microscopic model featuring it;
6 - The width of the gray-shaded area in Fig. 8 (now Fig. 9) has been corrected, and the agreement between the "transition region" and the onset of ergodicity in our finite-size data has somewhat improved. Nonetheless we already commented diffusely in our original manuscript about this agreement (which we termed as "reasonable"), and in particular on the fact that the entropy density s_Q produced by the quench dynamics might be underestimated, which in turns produces an overestimation of the injected energy density required to reach ergodicity. We do not find our claims to be particularly strong and "optimistic" on this topic - the strongest claim is that two distinct dynamical regimes exist, tuned by the quench amplitude, but we think that such a claim is rather objective in the face of our results;
7 - This is an interesting remark, although we are not sure in what sense our observations connect (or clash) with the estimate of a finite conductivity at finite temperature in contact with a heat bath. Here we are exploring the far-from-equilibrium unitary dynamics of the system in the absence of any heat bath, whereas the two references cited by the Referee focus on the linear response to an oscillating field for the system in thermal in equilibrium with a bath. The fact that the dc conductivity is always finite at any finite temperature in the latter system is per se not inconsistent with the subdiffusive behavior clearly observed in the unitary dynamics of the system we studied. Perfect insulation does not exist in disordered systems -- incoherent hopping assisted by the thermal bath is in principle always possible: see e.g. the famous variable-range hopping mechanism predicted by Mott;
8 - We were aware of this reference -- which goes along with a number of other ones concerning models with a quasi-periodic modulation of the parameters (either hopping or on-site potential or both). The article in question did not investigate the standard interacting Aubry-André model, but a variant thereof, which possesses a mobility edge already at the level of the single-particle spectrum. On the other hand we focused on a situation in which a mobility edge exists uniquely as a many-body effect, whence our choice of the standard AA model. Nonetheless we have added a reference to this work in the context of previous studies of the MBL transition via quenches from random factorized states.

---

## Round 1 · Referee Report · Anonymous (Referee 2) · 2016-9-20

Strengths

1-Systematic study of energy dependence after a global quench, revealing the many-body mobility edge
2-Clearly written and detailed

Weaknesses

1-Finite size scaling is not very systematically analyzed.

Report

This is a detailed study of the properties of the interacting quasi-periodic Aubry-Andre model, which shows a many-body localization (MBL) transition. Starting from a ground state phase diagram (obtained with a combination of Monte Carlo and DMRG) and the observation that reversing the sign of the interaction takes the ground state to the highest excited state, the authors argue that this model necessarily exhibits a many-body mobility edge, since the phase diagram is asymmetric in the interaction strength. That is, for a given set of parameters close to the MBL transition, the many-body spectrum consists of separate regions of extended and localized states. For small system sizes, accessible by exact diagonalization, this observation is corroborated by calculating standard measures of MBL---level spacing statistics, particle fluctuations and entanglement entropy---leading to a energy-density vs. potential strength phase diagram with a clear many-body mobility edge.

At this point, there is perhaps not much new apart from details, but these are useful details. My only criticism here would be that the extraction of critical disorder values seems to be done essentially by eye, and it's not clear that the error bars of this procedure are not much larger than what is given. For example, in Fig. 3, there is never a regime where the level spacing statistics in the localized regime actually gets all the way to the Poisson value. The level spacing statistics in this regime also depends only weakly on the system size, and it's not at all clear that the grey window accurately captures the proper uncertainty in where the transition actually takes place. This is important since the existence of the many-body mobility edge means that the transition depends on energy, and the error bars might be so large that it would be hard ton convincingly conclude from this data that the mobility edge is there. The same can be said about the data for the entanglement entropy. That being said, the data is reasonably good, and it is likely hard to do much better than this for energy densities close to the edges of the spectrum.

The main new element in this work is the systematic study of the energy density dependence of quantum dynamics after a global quench. The particular protocol that is chosen here is to chance the amplitude of the quasi-periodic potential, which is also experimentally possible and relevant, starting from the ground state at \Delta_i and quenching to \Delta_f. Depending on how much this amplitude is changed, the amount of energy density change compared with the ground state of the final Hamiltonian changes, and this is plotted in Fig. 6. One observes that it's possible to tune this energy density from being small (final energy close to the ground state energy) to being large (final energy close to the maximum energy state). Via this quench protocol the authors are able to systematically study the energy dependence of the dynamics and thereby observe the many-body mobility edge in dynamical properties.

The dynamical properties that the authors study are, like the eigenstate properties studies before, standard measure of MBL, namely entanglement entropy and particle number fluctuations. In the ergodic phase the entanglement increases quickly to the expected ergodic value, while in the localized phase it increases logarithmically to a sub-thermal value. The results that are obtained are what is expected and are consistent with the several prior studies that have studied these quantities. As noted above, the main contribution of this study is the systematic study of the energy dependence in the particular quench they look at.

Again here, as in the eigenstate study, I find the estimation of the critical disorder strength not to be very systematic---or at least it is not systematically and clearly explained. The comparison with the grey regions obtained from the level spacing statistic has overlap with peaks in the energy derivative of the entanglement and particle number fluctuations. However, looking at the energy level spacing data it seems equally reasonable that the transition is at somewhat higher energy densities, and then the agreement would seem coincidental.

In any case, a more systematic extraction of critical energy densities and error bar analysis would be nice, but it's unlikely to change things much qualitatively, but I think it will change the certainty with which one can state that there is a many-body mobility edge. The skeptics that don't think that the data obtained in prior work, seemingly demonstrating a many-body mobility edge, are perhaps not likely to be completely swayed be these results. The study is nevertheless solid and detailed and it is a good addition to the literature on many-body localization in the interacting Aubry-Andre model.

Requested changes

1-Explain in more detail how the estimates of the critical energies for the MBL transition is obtained (the grey regions in several plots) and give arguments for why this is a reliable estimate. If possible, present a more systematic finite size scaling analysis to further support the data presented.

  • validity: high
  • significance: ok
  • originality: ok
  • clarity: high
  • formatting: excellent
  • grammar: excellent

Author:  Tommaso Roscilde  on 2016-10-21  [id 69]

(in reply to Report 2 on 2016-09-20)
Category:
answer to question
reply to objection

We thank the Referee for the positive report, and for the useful remarks. We have revised all our estimates of the transition points and error bars estimated from the scaling of the adjacent-gap ratio, now obtained in a systematic way in the data reported on the spectral phase diagram, and correctly reflected by the width of the gray-shaded areas in the other figures. Several naive mistakes had been made in the previous version in both aspects, which are now removed. The criteria for estimating the transition point and its error bar are now explicitly stated in the text: they are based on the crossing points of the adjacent-gap ratio r among the four largest sizes we considered, namely L=16 with L=14, and L=15 with L=13 (only the crossing between sizes of the same parity is considered, given that even-odd effects are present in our results due to the half filling condition which is differently realized in the two cases). The error bar simply reflects the spreading of these crossing points, and it therefore identifies the region over which we observe an inversion of scaling for the four largest system sizes. The width of all the shaded areas reported in the figures now reflects the error bar estimated in this way -- we have modified Figs. 3, 6, 7, 9 and 11 in this sense.
We believe that above criterion leads to a rather honest and conservative estimate of the error bar. Even within this estimate of the error bar, the extended-MBL transition line appears to be clearly dependent on the energy density, revealing the existence of a mobility edge. The fact that the data in Fig. 3 never reach the Poisson value is easily understood in that, unlike in previous studies, here the r ratio is plotted as a function of the energy density at *fixed* strength of the quasi-disordered potential \Delta, something which does not allow the localisation length to become arbitrarily small, and, in particular, much smaller than the system size: under this condition, it should not be possible to observe Possonian statistics in the proper way. We have added an explicit remark in the text. On the other hand, as shown in the false-color part of the spectral phase diagram (now Fig. 5 of the new version) there are \Delta values (such as \Delta ~ 3) for which the excursion from the level-repulsion statistics to the Poissonian statistics is better realized. To be even more convincing in this sense, we have added to our manuscript a new figure (Fig. 4) containing the scaling of the r ratio as a function of \Delta for two different values of the energy density. Working at variable \Delta allows to approach arbitrarily well the Poisson-statistics value for r. In particular, this figure shows rather clearly that the crossing regions between the various r curves (estimated as declared above) are non-overlapping for different energy densities, witnessing again the existence of a mobility edge.
Coming to the data concerning the dynamics, we do not fully understand the remark of the Referee concerning the the overlap between the peaks in the energy derivative of the entanglement and fluctuations, and the estimate of the transition region from the level statistics -- the latter is an analysis which pertains uniquely to the study of the spectral properties. We disagree with the idea that the correspondence between the two estimates of the transition could be coincidental, given the clear trend that they both follow in the (\Delta,\epsilon) plane in Fig. 5. The estimate of the mobility edge stemming from the dynamics is shown in Figs. 9 and 11, both for what concerns the long-time dynamics (Fig. 9) as well as the short-time one (Fig. 11). It seems hardly coincidental to us that full ergodicity at long times is only observed when the injected energy density clearly exceeds the mobility edge estimated from level statistics, and that e.g. the logarithmic-in-t onset of particle number fluctuations appears only once that same threshold has been exceeded.

---

## Round 3 · Referee Report · Anonymous (Referee 1) · 2016-10-21

Strengths

See previous report

Weaknesses

See previous report, the authors have satisfactorily addressed these points.

Report

With the revisions implemented in the new version, the authors have, in my opinion, satisfactorily addressed both Referees' comments. I recommend the publication of this very nice ,well written and relevant publication as is.

Just two clarifications/comments on the authors' response (with no intention to
trigger additional revisions):

Ad 5) An inverted mobility edge was predicted by Altshuler et al. in Phys. Rev. Lett. 113, 045304. Moreover a 2D Bose-Hubbard model with disorder at for the parameters of Ref. 14, is in a superfluid state at T=0 (see work by Svistunov, Prokoviev, Pollet and others). Given that this experiment seems to observe many-body localization at higher energy densities, such an inverted mobility edge could exist.

Ad 7) Here I primarily wanted to point out that while there is ample evidence
for subdiffusive dynamics in a portion of the ergodic phase (be it based on
equilibrium or far from equilibrium probes), there is nevertheless an inconsistent picture when it comes to computing the conductivity itself.

I disagree with the authors' perception of the role of baths in the calculation of conductivities. As far as I understand, conductivities are bulk properties and transport is an intrinsic property, the bath just provides temperature and perhaps other control parameters to define the appropriate equilibrium ensemble, but it is (in the present context) not the source of conducting behavior per se.
In linear response, the claim of subdiffusive dynamics and a finite dc conductivity seem irreconcilable to me. Recall that the very phenomenological definition of the many-body localized phase goes via sigma_dc=0 (computed in a thermal ensemble!), see Basko, Aleiner and Altshuler.
~

Requested changes

1- The authors could perhaps update Refs. 14,15, & 55, which in the meantime all were published.

---

## Round 3 · List of Changes

The criterion for the estimate of the MBL-extended transition region from the scaling of the adjacent gap ratio is thoroughly discussed and implemented in section 3.1. A sentence is added in the conclusions concerning the possible experimental signatures. Fig. 4 has been added, the newly obtained estimates of the transition from the adjacent-gap ratio are reported in Fig. 3, and an updated gray-shaded area -- marking the transition -- appears now in the modified version of Figs. 5, 6, 7, 9, 11.

---

## Editorial Decision

published